# Exercise Dosage in Reducing the Risk of Dementia Development: Mode, Duration, and Intensity—A Narrative Review

**DOI:** 10.3390/ijerph182413331

**Published:** 2021-12-17

**Authors:** Sukai Wang, Hong-Yu Liu, Yi-Chen Cheng, Chun-Hsien Su

**Affiliations:** 1College of Physical Education, Huaqiao University, Quanzhou 362021, China; wsk1019@163.com; 2Department of Exercise and Health Promotion, Chinese Culture University, Taipei 111369, Taiwan; lhy2@ulive.pccu.edu.tw (H.-Y.L.); cyz29@ulive.pccu.edu.tw (Y.-C.C.); 3College of Kinesiology and Health, Chinese Culture University, Taipei 111369, Taiwan

**Keywords:** exercise dosage, exercise mode, exercise duration, exercise intensity, physical activity, Alzheimer’s disease

## Abstract

Senile dementia, also known as dementia, is the mental deterioration which is associated with aging. It is characterized by a decrease in cognitive abilities, inability to concentrate, and especially the loss of higher cerebral cortex function, including memory, judgment, abstract thinking, and other loss of personality, even behavior changes. As a matter of fact, dementia is the deterioration of mental and intellectual functions caused by brain diseases in adults when they are mature, which affects the comprehensive performance of life and work ability. Most dementia cases are caused by Alzheimer’s disease (AD) and multiple infarct dementia (vascular dementia, multi-infarct dementia). Alzheimer’s disease is characterized by atrophy, shedding, and degenerative alterations in brain cells, and its occurrence is linked to age. The fraction of the population with dementia is smaller before the age of 65, and it increases after the age of 65. Since women live longer than men, the proportion of women with Alzheimer’s disease is higher. Multiple infarct dementia is caused by a cerebral infarction, which disrupts blood supply in multiple locations and impairs cerebral cortex function. Researchers worldwide are investigating ways to prevent Alzheimer’s disease; however, currently, there are no definitive answers for Alzheimer’s prevention. Even so, research has shown that we can take steps to reduce the risk of developing it. Prospective studies have found that even light to moderate physical activity can lower the risk of dementia and Alzheimer’s disease. Exercise has been proposed as a potential lifestyle intervention to help reduce the occurrence of dementia and Alzheimer’s disease. Various workout modes will be introduced based on various physical conditions. In general, frequent exercise for 6–8 weeks lessens the risk of dementia development.

## 1. Introduction

Dementia is not a single disease, but a group of symptoms (syndrome). Symptoms include memory loss, and it can affect other cognitive functions, including deterioration of language, spatial sense, computation, judgment, abstract thinking, attention, and other functions [1]. Symptoms such as disruptive behavior, personality changes, delusions or hallucinations may also occur, which are severe enough to affect interpersonal relationships and the ability to work [2]. The linkage of physical activity/exercise and dementia is strong. We will demonstrate the connection in the second section below. In the classification of dementia, there are roughly two categories: degenerative and vascular, but patients sometimes have two or more causes, the most common is Alzheimer’s disease and vascular dementia coexist (also known as hybrid) [3,4]. The above two types of dementia are articulated as follows:

### 1.1. Degenerative Dementia

Most patients fall into this category, among which the following three are the most common.

(1)Alzheimer’s disease

It is the most common form of dementia, named after a German doctor, Alois Alzheimer, who discovered it in 1906. In the early stage, the most obvious symptoms are memory decline, problems in the identification of time, place, and people, and more than two kinds of cognitive dysfunction, which is a progressive degradation and irreversible. It is a neurodegenerative disease in which nerve cells in the brain are damaged [5]. Doctors judge through computer tomography and nuclear magnetic resonance that Alzheimer’s disease mainly affects the hippocampus in the early stage, and abnormal senile plaques and nerve fiber tangles can be found in the posterior brain anatomy. The clinical course is about 8–10 years [6].

(2)Frontotemporal lobe degeneration

Brain disorders mainly invade the frontal and temporal lobes, characterized by early personality changes and loss of behavioral control, with patients often having irrational behavior [7]. Alternatively, patients suffer from early onset of language impairment, such as dysphoria, and other gradual declines that occur on average after the age of 50 [8].

(3)Dementia with Lewy bodies

As the second most common degenerative dementia, in addition to the cognitive dysfunction, in early stage, patients may experience stiffness, hand shaking, walking instability, and repeatedly unexplained fall phenomenon. There are also obvious mental symptoms, such as distinct visual or auditory hallucinations mood swings or suspicious delusion symptoms [9].

(4)Others include dementia caused by Huntington’s disease and so on

Neurodegenerative dementia is considered as a proteomatosis characterized by the accumulation of specific proteins in the brain, such as beta-amyloid in AD and the microtubule-associated protein tau or alpha-synuclein in Lewy body dementia [10]. Although drugs have not been proven to have a neuroprotective effect on dementia, the growing body of literature demonstrates that long-term, regular exercise has significant benefits on cognition, dementia risk, and dementia progression [11,12].

### 1.2. Vascular Dementia

Vascular dementia is due to stroke or chronic cerebrovascular lesions, resulting in poor blood circulation in the brain. Brain cell death caused by mental decline is the second leading cause of dementia [13]. Generally, vascular dementia occurs after stroke [13]. If a stroke victim survives, roughly 5% of survivors will develop dementia symptoms. Dementia occurred in 25% of the cases after five years of surveillance. [14]. Its characteristics are sudden deterioration of cognitive function, ups and downs, step-by-step degradation, slow movement, slow response, gait instability and mental symptoms [13]. Primary vascular dementia is common, and cerebrovascular small vessel diseases too (e.g., white matter osteoporosis and lacunar disease) seem to combine with neurodegenerative processes to cause dementia [15]. These atherosclerotic cerebrovascular mechanisms differ from neurodegeneration and age-related loss of neurofibril networks and synapses [13,16]. Exercise can reduce the effects of atherosclerotic cerebrovascular disease to promote brain health. Since the benefits of exercise in reducing the risk of atherosclerosis (cerebrovascular disease) are well established, exercise may have a more immediate beneficial effect on brain neuroplasticity and resilience to brain aging and neurodegeneration [11,17].

Case–control studies have suggested that increased physical activity can significantly reduce the risk of dementia in older people [18]. In addition, past studies, from the physiological and molecular biological levels of nerve cells, have provided favorable evidence that increased physical activity can prevent dementia in older people [19]. Studies have shown that hydrogen peroxide oxidation(H_2_O_2_) damage in the body can cause the accumulation of amyloid beta-amyloid (Aβ) and alpha-synuclein (α-syn) in the cells of normally inactive people, which is not the case in endurance athletes [20,21,22]. Systematic review and meta-analysis studies confirm that the effects of exercise prevention of various types of dementia and Alzheimer’s disease show a dose response trend. the greater the amount of physical activity, the better the prevention effect [12].

The decline of the brain’s hippocampus and prefrontal cortex, which play important roles in memory formation and complex thinking, is a risk factor for Alzheimer’s disease. Surprisingly, these are the same areas that respond to physical activity. Exercise enhances brain growth and prevents cognitive decline. According to a study, older adults who were physically active, even if it was low-intensity activity such as gardening or golf, had larger brains compared to those inactive seniors [18].

## 2. The Linkage of Physical Activity/Exercise and Dementia

Regular physical exercise appears to be one of the effective methods to reduce the risk of dementia among all the lifestyle changes that have been studied [23]. Several studies on the effect of aerobic exercise (exercise that raises your heart rate) in middle-aged or older adults have found improvements in thinking and memory, as well as a reduction in dementia rates [24].

The definition of ‘physical activity’ or exercise used in research studies in sport science area varies [25]. The American College of Sport Medicine (ACSM) defines physical activity as any bodily movement produced by skeletal muscle contraction that results in a significant increase in caloric requirements over resting energy expenditure [26]. Exercise is a type of physical activity that consists of planned, structured, and repetitive bodily movement that is performed to improve and/or maintain one or more components of physical fitness [27].

### 2.1. Increased Physical Activity Can Prevent Cognitive Decline

Epidemiological generational follow-up studies, case–control studies and systematic retrospective studies have all shown that the risk of Alzheimer’s disease, vascular dementia or overall dementia is reduced, regardless of the amount of physical activity, the amount of physical activity during leisure, the amount of total physical activity or the better level of fitness [11,28,29]. Increasing physical activity among the elderly has the effect of preventing dementia, and it also has the effect of improving dementia with the help of psycho-emotional aspects such as depression and behavioral and psychological symptoms of dementia (BPSD) [30].

Physical activity may reduce the age-related risk of all-cause dementia in older adults, according to the results of a population-based cohort study presented at the Alzheimer’s Association International Conference. The study analyzed data from 8270 participants classified as inactive, low, or moderate-to-vigorous physical activity. Results showed that 7.8% of participants were diagnosed with all-cause dementia. The risk of all-cause dementia increased by 6.1% with age. Participants who engaged in low- or moderate-to-high-intensity physical activity had a reduced risk of all-cause dementia. People aged 80 or older who engaged in moderate to vigorous physical activity had a lower risk of all-cause dementia than inactive adults aged 50 to 69 [31].

People should exercise to mitigate the negative effects of aging on their cognitive function [32]. Physical activity and exercise may slow cognitive decline. The results of several recent randomized controlled trials (RCTS) suggest that physical activity and exercise have positive effects on cognition of people with cognitive decline. The results show that physical activity and exercise can improve the cognitive performance of people with cognitive impairment to some extent [33,34,35], but it is not clear which combination of frequency, intensity, duration, and type of exercise has a better effect on cognitive performance. To find the best interventions, there is an urgent need to answer the question of how to conduct randomized controlled trials of physical activity and exercise to improve cognition of exercise dosages. The study of the dose–response effect of physical activity on cognition of AD patients is a problem that needs to be solved.

### 2.2. Increasing Physical Activity Can Improve Psycho-Emotional Aspects

In addition to cognitive dysfunction, dementia is often accompanied by emotional, behavioral, psychiatric symptoms and other “non-cognitive symptoms” [36]. These symptoms can cause more discomfort for patients and are a major source of stress for caregivers. “Non-cognitive symptoms” include depression, delusions, confessions, hallucinations, and other mental behavioral disorders [36,37]. Up to 70 to 80 percent of dementia patients develop “non-cognitive symptoms” during their illness.

Different types of dementia are associated with different behavioral and psychiatric symptoms. 1. Alzheimer’s disease—common symptoms include apathy, depression, anxiety, delusions, agitation, and impatience [38,39]. 2. Frontotemporal lobe degeneration—the most obvious symptoms are impulsiveness, doing the same thing repeatedly and profanity. They have no sense of sickness. They demonstrate an inability to control behavior and language, and loss of normal social functions at an early age [40]. 3. Dementia with Lewy bodies—there is a higher proportion of behavioral and psychiatric symptoms, about 80% of patients with vivid visual hallucinations, patients often insist that there are some people at home they do not know, and even talk to them. Not only do they suffer from insomnia, but they may also have REM sleep behavior problems [41].

Exercise helps the brain release endorphins, chemicals that make people feel happy, and studies have shown that regular exercise makes people feel happy for a long period [42]. A study from The Lancet, the world’s oldest medical journal, found that those who exercised regularly had 1.5 fewer depressed days per month than those who did not exercise. It also suggested that 45 min of exercise, three to five times a week provided the greatest benefits for mental health [43]. Michael Miller, a Harvard Medical School PhD, also wrote in a separate article that exercise helps the brain’s hippocampus grow, which helps ease depression [44]. In addition to endorphins, many people experience a significant stress relief after exercise because exercise reduces the production of cortisol, the “stress hormone”, and adrenalin, which can cause anxiety [45]. Previous studies also found that strength training significantly reduced anxiety symptoms [46,47]. There are several studies that consistently show that exercise leads to greater happiness in both the short and long term, and that increasing physical activity can even improve the mood of people with bipolar disorder the day after exercise, which produces serotonin and makes them feel good [48,49]. Studies have also shown that the increase in body temperature during exercise increases blood circulation in the brain and affects the HPA axis (hypothalamic–pituitary–adrenal axis) in the brain, which helps improve mood [50]. In addition, there are many noncognitive, nonvascular degenerative benefits that older people can derive from regular exercise. These include a reduced risk of osteoporosis and fractures, a reduction in age-related sarcopenia, as well as benefits for depression and anxiety. Exercise programs can improve behavioral management of dementia and fall risk [51,52,53,54,55,56]. Compared with indoor exercise, exercising outdoors in a natural setting can be energizing, reducing stress, anger and depression, and increasing energy levels, researchers say [57]. In addition to improving mood, exercise also improves the brain and central nervous system [57]. Researchers at the University of Texas at Austin have shown that high-intensity exercise improves cognitive performance [58], and another study has shown that moderate-intensity aerobic exercise improves short-term memory [59]. A study at the Georgia Institute of Technology found that just one cycle of strength training already improved memory over 48 h [60], and another study showed that aerobic exercise had the same effect [61].

## 3. Exercise Mode in Reducing the Risk of Dementia Development

Aerobic (i.e., endurance training), resistance (i.e., strength training), flexibility, and balance are the four main types of exercise training, which may be referred as exercise mode. All people should engage in a variety of physical activities to increase the components of physical fitness. Distinct types of physical activities target different aspects of physical fitness that are linked to health.

Physical activity has been shown to reduce the incidence of dementia; however, previous cohort studies have rarely looked at the various types of physical activity and household activities [62]. Physical activity was linked to a lower incidence of dementia in several longitudinal cohort studies, but not all of them [63,64,65]. Previous studies have been chastised for failing to evaluate the contribution of household activities and for not studying the different types of physical activity [66,67]. Studies have established a link between leisure-time physical exercise and the risk of all-cause dementia or Alzheimer’s disease (AD), with the AD risk curve tending to flatten at higher levels of activity [68]. As a result, it is crucial to investigate the risk function’s shape and discover the activity level thresholds that correspond to risk changes.

The inverse association between a physically active lifestyle and the risk of cognitive decline is well established, and aerobic exercise training has been the most widely utilized method for investigating the role of physical activity in reducing the detrimental effects of aging on cognitive performance [69].

The various workout modes listed below can be performed alone or in combination with other modes.

Aerobic exercises, also known as cardio workouts, are exercises that require movement. Aerobic exercise (also known as endurance activities, cardio, or cardio-respiratory exercise) is a type of low- to moderate-intensity physical activity that relies heavily on the aerobic energy-generating process. This type of exercise is beneficial to the heart and lungs. These workouts help to strengthen the heart and increase lung capacity. Walking, running, dancing, basketball, biking, kickboxing, step-ups or step aerobics, and swimming are the most prevalent forms of these exercises.Muscle development exercises include anaerobic exercises, resistance training, and strength training. Anaerobic exercise is a type of exercise that uses no oxygen to break down glucose in the body. Anaerobic means “without oxygen.” In practice, this means that anaerobic exercise is more intense than aerobic exercise but lasts for a shorter time. This type of exercise aids in the development of specific muscle groups. The advantages of this workout include increased muscle strength, size, and endurance. Dumbbell exercises, weightlifting, and using gym equipment are the best examples of these activities.Stretching is performed whenever people experience muscle strain. Stretching is a type of physical activity in which a specific muscle or tendon (or muscle group) is flexed or stretched to improve the muscle’s perceived flexibility and attain comfortable muscular tone. Stretching enhances the general flexibility of the body. It may also help to relieve muscle tension and cramps. Yoga, for example, is a methodical stretching and breathing regimen that is good to both the mind and body. It eases anxiety and muscle tension while also expanding your range of motion.Calisthenics, also known as Body Conditioning Workouts, are workouts that do not require the use of any equipment. Calisthenics is a type of strength training that consists of a series of motions that target major muscle groups while using the body’s own weight as resistance. The most well-known exercises for this sort of exercise are sit-ups, push-ups, squats, and lunges.Plyometrics are high-intensity workouts which are ideal for athletes and advanced exercisers, as well as exercisers with a well-conditioned body. Plyometrics are exercises that require muscles to exert maximum force for brief periods of time to increase power (speed–strength). Jumping rope, jumping jacks, throwing, and catching a ball against a wall, and boxing with a punching bag are the most typical workouts for this sort from regular exercisers.

Exercise intervention to prevent dementia requires selecting appropriate exercise patterns based on individual health and physical condition, as well as adhering to the principles of progressive load, overload, and specificity of exercise training. Targeting previous studies, exercise mode, findings and highlights are listed in Table 1.

According to the findings of these studies, exercise intervention to reduce dementia or Alzheimer’s disease has positive implications for lowering the risk of dementia or Alzheimer’s disease, with exercise patterns ranging from household activity to prescriptive exercise.

## 4. Exercise Duration in Reducing the Risk of Dementia Development

In 2018, the US Department of Health and Human Services proposed a physical activity key guideline for adults and older adults. The department has specific key recommendations for physical activity duration [26]. Adults and older adults should engage in at least 150 min (2 h and 30 min) of moderate-intensity aerobic physical activity per week, or 75 min (1 h and 15 min) to 150 min (2 h and 30 min) of vigorous-intensity aerobic physical activity per week, or an equivalent combination of moderate- and vigorous-intensity aerobic activity. Aerobic activity should ideally be spread out over the course of the week. Adults and older adults who are unable to engage in 150 min of moderate-intensity aerobic activity per week due to chronic conditions should be as physically active as their abilities and conditions allow [73].

Several previous studies have looked at physical activity and dementia, as well as the relationship between time spent physically active and dementia. Najar and colleagues created a study about cognitive and physical activity and dementia in which subjects were divided into four groups. Group 1 was completely inactive, spending most of its time watching television and going to the movies. Group 2 engaged in light physical activity for at least 4 h per week, such as walking, gardening, bowling, or cycling for 30 min per day. Group 3 engaged in regular physical activity such as running, tennis, or swimming for at least 2–3 h per week. Group 4 received regular–intense physical training, such as heavy exercise, such as running or swimming several times per week, or participation in competitive sports [74]. They discovered that physical activity duration in midlife was associated with a lower risk of mixed dementia and dementia with cerebrovascular disease [74]. A meta-analysis study on the effects of physical activity and exercise on the cognitive function of patients with Alzheimer’s disease revealed that all interventions were conducted for an average of 40 min per session, ranging from 30 to 60 min. Interventions lasted at least one hour per week. The total duration ranges from 12 to 24 weeks, with a mean intervention duration of 16.92 weeks. This meta-analysis study, which compared different amounts of physical activity and exercise interventions for Alzheimer’s disease in detail, suggested that physical activity and exercise can improve cognition in older adults with Alzheimer’s disease [12]. Maass and colleagues investigated the effect of physical activity on cognitive performance in healthy older people and found no significant link between physical activity and cognitive functions. In comparison to other studies, this one only used a 30 min exercise duration; all other studies used physical activities that lasted between 60 and 90 min. The length of the experimental interventions and the duration of the exercise may have played an even larger role in this case. As a result, it can be assumed that 30 min of exercise (despite the same number of training units per week) may be deemed insufficient in terms of impact on cognitive functions [22].

Previous research has measured physical activity in a variety of ways, examined various types and intensities of activity, and employed a variety of exercise duration and frequency. It has not been possible to provide an optimal exercise prescription for brain health and function, or to reduce the risk of dementia. The good news is that these findings suggest that many types of physical exercise and physical activity are beneficial to brain health and cognitive function when taken together. Table 2 summarizes studies on exercise intervention in dementia in terms of exercise training duration.

Non-resistance exercise training typically lasts between 30 and 60 min per exercise intervention, whereas resistance exercise training lasts about an hour per exercise intervention, according to research [71,75,76,77,78,79,80].

## 5. Exercise Intensity in Reducing the Risk of Dementia Development

The amount of energy expended when exercising is referred to as exercise intensity. Heart rate is commonly used to determine the intensity of exercise. Heart rate can be an indicator of the cardiovascular system’s challenge that exercise represents. The most difficult challenge in designing an exercise program is determining the appropriate exercise intensity. HR and rating of perceived exertion (RPE) are the two most used methods for prescribing and monitoring exercise intensity. HR is used to set an exercise intensity range because there is a linear relationship between HR and percent functional capacity VO_2_. For the younger population, exercise intensities of 60–80 percent are generally recommended. In the elderly, however, an exercise intensity of 40% of HR reserve has demonstrated aerobic and functional training adaptations [81].

The Dementia and Physical Activity (DAPA) trial was commissioned by the National Institute for Health Research (NIHR) to inform the debate about the potential benefit of exercise on cognitive impairment in people with dementia [82]. A previous study used longitudinal data to examine the physical activity behavior and cognitive function of 16,700 Europeans aged 54 to 75 over a 13-year period. They discovered that moderate physical activity on a weekly basis has a potential and direct protective effect against cognitive decline and dementia, with women benefiting more than men. A brisk walk is an example of moderate physical activity, whereas running or circuit training is an example of vigorous physical activity [83]. One study, published in the journal *Applied Physiology, Nutrition, and Metabolism*, in 2019, looked at the link between workout intensity and memory enhancement. Researchers divided a group of 64 elderly participants into three groups: one that did high-intensity interval training, one that did moderate continuous training, and one that did not do any training. While high-intensity interval training improved memory the most, researchers concluded that exercise intensity did not affect executive functioning because positive trends were observed in both exercise groups. The study discovered that general fitness improvement correlated with improved memory performance [84]. Researchers suggest that intensity is critical. Seniors who exercised using short, bursts of activity saw an improvement of up to 30% in memory performance while participants who worked out moderately saw no improvement, on average [84]. Eleven young men (average age of 25) and ten older men (average age of 69) participated in the study, which was published in the journal *Medicine & Science in Sports & Exercise*. The researchers discovered that during interval exercise, there was a greater overall change in the total accumulated volume of blood flow over the duration of both exercise and recovery for both populations than during steady-state exercise. Brain blood flow declines gradually with age and has been linked to the risk of cognitive decline, including dementia. These increases in brain blood flow caused by interval exercise could be beneficial to future brain health [85]. The basis indicators of exercise intensity vary in the research design of an exercise intervention to reduce dementia, ranging from moderate to high intensity. Table 3 organizes the relevant research findings.

The exercise intensity in the early studies on the exercise intervention for dementia was mostly set to moderate. In recent years, there has been a gradual increase in high-intensity exercise intervention or high-intensity interval exercise intervention, with the exercise intensity being 70–75% of the maximum heart rate, 80% of the maximum oxygen consumption, and 60–75% of the heart rate reserve [84,86,87,88,89,90,91].

## 6. Maintain Physical Activity and Exercise Early May Prevent Dementia

Several studies have found that inactive people have smaller brains, poorer cognitive function, and a decline in structural brain health as they age [92,93]. Experts call for good habits to be formed from the early 20 s to prevent rapid brain degeneration [93].

### 6.1. Homeostasis

Homeostasis is an important concept that summarizes all aspects of aging from a functional standpoint. Homeostasis refers to the physiological processes that keep the body’s internal environment stable [94]. Individuals’ ability to tolerate stressors declines as they age, but it is partially modifiable through lifestyle changes. Mueller and Maluf’s physical stress theory (PST) capture the essence of homeostasis [95]. The successfully aging older adult has a high tolerance to physiological stressors that challenge homeostasis, whereas the unsuccessfully aging older adult has a low tolerance to physiological stressors that challenge homeostasis. Tolerance range rises in response to exercise and falls in the presence of chronic disease and increased inactivity [94,96]. Exercise results in a strong positive change with systemic adaptation when a person is in homeostasis. Strength and balance, as well as aerobic and muscle endurance, can improve. Positive change occurs when an inactive older adult with stable chronic diseases engages in exercise, albeit more slowly and to a lesser extent [94,97,98].

### 6.2. Body Composition

Over decades, there is a gradual shift in body composition in which lean mass decreases and fat mass increases proportionately, for example, an aging male with a typical shift in fat and lean mass. After the third decade, lean mass, which is mostly muscle, continues to decline. Concurrently, fat mass increases. Body weight has not changed in this individual over the 60 years represented [94,99,100]. The fact that most of the fat increase occurs inside the peritoneum, which is now thought to be a significant contributor to the increased inflammation that occurs with age, is significant. Increased intra-abdominal fat is also thought to predispose older people, especially women, to elevated lipids and prediabetes [101]. Fat is a highly active metabolic tissue, and its role in age-related decline and disease is only beginning to be understood. Men and women of all ages who are physically active on a regular basis do not accumulate intra-abdominal fat to the same extent as those who are sedentary [102]. Consequently, active men and women have less whole-body inflammation and less disease [103]. Previous research investigated the relationship between fitness and dementia risk, and the results of brain MRI and cognitive tests revealed that inactive people performed worse on smaller brain volume tests at the age of 60. It was hypothesized that prolonged sitting would reduce brain neurogenesis, angiogenesis, and synaptic plasticity, as well as increase inflammatory responses, all of which would have an impact on brain health [104,105]. Obesity is middle age 45 to 60 years old suffering from Alzheimer’s disease, one of the risk factors [43]. The accumulation of abdominal fat reduces brain volume, increasing the risk of Alzheimer’s disease in the future. Because visceral fat in the abdomen produces more inflammatory cytokines. They are also more likely to have hypertension, diabetes, and cardiovascular disease, which can all lead to vascular brain atrophy and have a negative impact on brain health and cognitive function.

### 6.3. Cardiovascular Tissues

Fundamental changes in vascular tissues that occur with aging are summarized as follows: a decrease in maximum heart rate, a decrease in VO_2_max, stiffer, less compliant vascular tissues, loss of cells from the SA node, decreased contractility of the vascular walls, and thickened basement membrane in capillaries. The most noticeable and clinically significant change is the decrease in maximum heart rate [106,107]. The typical formula of 220 minus age provides a rough guideline for the expected change in maximum heart rate. Thus, an 80-year-old person is likely to have a maximum heart rate (HRmax) of 140 bpm, limiting the extent of cardiovascular challenge that can be tolerated for any length of time [108]. There is a link between muscle mass and VO_2_max, which is why men have higher max values than women. At any age, the greater the lean mass, the greater the maximal aerobic capacity. Sarcopenia is characterized by a very low aerobic capacity [109,110]. Inactivity has been linked to being overweight or obese, type 2 diabetes, certain types of cancer, and premature death in studies. Too much sitting in general, as well as prolonged sitting, appears to increase the risk of death from cardiovascular disease [111]. People with regular exercisers are less likely to suffer from heart disease and stroke, both of which are linked to an increased risk of dementia. Physical activity is also important for maintaining adequate blood flow to the brain and may promote brain cell growth and survival. Exercise is thus one of the factors being researched for its role in lowering the risk of developing dementia as well as the benefits it provides to people with dementia [112,113].

### 6.4. Nervous System

Fundamental changes in the central and peripheral nervous systems have significant implications for function. The following are some of the major age-related changes in the nervous system: Myelin sloughing/loss Slowed nerve conduction, axonal loss, autonomic nervous system dysfunction, sensory neuron loss, and slowed response time are all symptoms of slowed nerve conduction (speed of reaction) [114]. The balance of parasympathetic and sympathetic nervous system output changes with age, which is likely related to slowing gastric motility, bladder control issues, hypertension and hypotension, and deficits in control of blood flow to and from the periphery [115]. In an early study, researchers at the University of Maryland discovered that treadmill walking a 12-week program changes the ability of the brain involved in semantic memory movement after 4 months, the brain in the process of semantic memory test becomes less active, because they can use less response to deal with information. The findings suggest that exercise improves the brain’s ability to process semantic memories [116]. According to a study published in *Frontiers in Aging Neuroscience*, normally healthy older adults who stop exercising for only 10 days or so have brain regions responsible for physical thinking and learning. Blood flow to memory-related areas significantly decreases [117].

In addition to the gradual loss of physical strength and bone mass after the age of 50, the brain shrinks at a rate of 0.5 percent per year beginning at the age of 60 [118]. Continued shrinkage may increase the risk of cognitive impairment or dementia in the future. Many studies have shown that lifestyle patterns in childhood and adolescence influence future brain health, with inactive people having smaller brains, poorer cognitive function, and worse structural health as they age [92,118]. A UCLA study of 35 non-dementia adults aged 45 to 75 published in the journal PLOS ONE in 2018 discovered that prolonged sitting was associated with decreased thickness of the medial temporal lobe, which is responsible for the hippocampus and memory function [104]. It is speculated that prolonged sitting may reduce neurogenesis, angiogenesis, and synaptic plasticity in the brain, and increase inflammation, which affects the health of the hippocampus [119]. A 2015 Boston University study of 1271 adults aged 40 who were followed for 20 years found that inactive people had smaller brains and poorer test results by the time they were 60, using brain MRI and cognitive tests [120]. Furthermore, in a 2016 study published in *JAMA Psychiatry, the Journal of the American Medical Association*, 3247 adults aged 18 to 30 were followed for 25 years using three tests to assess cognitive functions such as verbal memory, executive function, and reaction speed, compared to those who exercised more and watched less television [121]. People who exercised less and sat in front of the television for more than three hours per day were more than twice as likely to have poor cognitive function later in life [122]. It is becoming clear that a healthy lifestyle that includes regular exercise can have a significant impact on preventing physical decline and disease. Regular exercisers (at any age) have lower rates of cardiovascular disease, osteoarthritis, diabetes, vascular disease, metabolic syndrome, pain, and Alzheimer’s disease, to name a few.

## 7. Conclusions

Dementia is a type of brain disease that causes a gradual, long-term decline in thinking and memory, impairing a person’s ability to function in daily life. Dementia affects 36 million people worldwide, with approximately 10% developing it during their lifetime.

Dementia can be prevented, and there is a growing body of research demonstrating this. All studies emphasize on changing eating and exercise habits, as well as increasing mental training. Exercise has been shown effective to both prevent and delay the progression of dementia. Aerobic exercise and muscle-strengthening exercises are preferable. Exercise may have a preventive effect due to the neurotrophic factor (BDNF) secreted by the brain, which can prevent hippocampal atrophy and maintain cognitive function. However, because everyone’s physical condition differs, the need for preventive measures should differ as well.

The human brain is a very complicated organ. It requires a certain intensity of regular exercise to effectively prevent dementia disease and to achieve real improvement in neural cognitive function and related blood biochemical indexes. The combination of aerobic activity and resistance exercise has a better effect on preventing dementia according to recent abundant researches.

## Figures and Tables

**Table 1 ijerph-18-13331-t001:** Exercise mode for dementia prevention.

Title	Cohort	Mode	Findings	Highlights	Reference: Grade
Physical activity types and risk of dementia in community-dwelling older people: the Three-City cohort	*n* = 1550 median age = 80 female participants = 63.6%	household/transportation activities	dementia was significantly and negatively associated	importance of considering all physical activity types in 72 years or older people	[62]: B
		leisure and sport activity	no associated		
Physical activity, cognitive decline, and risk of dementia: 28-year follow-up of Whitehall II cohort study	*n* = 10,308 age = 35–55 participants = 6895 men and 3413 women	mildly energetic (e.g., weeding, general housework, bicycle repair)	physical activity does not affect the risk of dementia	1. Changes in physical activity could just be part of the dementia’s preclinical symptoms. 2. There is no indication that persons who engage in more physical activity have a reduced rate of cognitive deterioration.	[63]: B
		moderately energetic (e.g., dancing, cycling, leisurely swimming)	physical activity does not affect the risk of dementia		
		vigorous physical activity (e.g., running, hard swimming, playing squash)	physical activity does not affect the risk of dementia		
Physical activity, diet, and risk of Alzheimer disease	*n* = 1880 mean age = 77.2	light (walking, dancing, calisthenics, golfing, bowling, gardening, horseback riding)	no significant	more physical activity is associated with a reduction in risk for developing AD	[70]: B
		Moderate (bicycling, swimming, hiking, playing tennis)	some physical activity had a 25% to 38% lower risk for AD		
		vigorous (aerobic dancing, jogging, playing handball)	much physical activity had a 33% to 48% lower risk for AD		
Exercise Is Associated with Reduced Risk for Incident Dementia among Persons 65 Years of Age and Older	*n* = 1740 mean age = 73.2 (free of dementia) mean age = 78.2 (with dementia) mean age = 76.3 (died or withdraw)	walking, hiking, bicycling, aerobics or calisthenics, swimming, water aerobics, weight training or stretching, or other exercise.	Physical exercise and performance-based physical function was statistically significant	physical exercise is associated with delayed onset of dementia or Alzheimer disease	[71]: B
Leisure-time physical activity at midlife and the risk of dementia and Alzheimer’s disease	*n* = 1449 in 1972 age 65–79 years participated in the re-examination in 1998 (mean follow-up, 21 years)	leisure-time physical activity	Leisure-time physical activity at midlife at least twice a week was associated with a reduced risk of dementia and AD.	Leisure-time physical activity at midlife is associated with a decreased risk of dementia and AD later in life.	[72]: A
The Relationship Between Physical Activity and Dementia: A Systematic Review and Meta-Analysis of Prospective Cohort Studies	none	Vigorous exercise, regular exercise, leisure time physical activities, and gardening	positive effect toward lowering dementia risk	Participation in physical activities produces a favorable effect toward lowering dementia risk.	[67]: D
Leisure time physical activity and dementia risk: a dose–response meta-analysis of prospective studies	none	leisure time physical activity (LTPA)	all-cause dementia or AD exhibited a linear relationship with LPTA	the dose–response relationship between LTPA and dementia, further supporting the international physical activity guideline from the standpoint of dementia prevention	[68]: D

The level of evidence of each reference is graded as A: overwhelming data from randomized controlled trials (RCTs); B: quasi-experimental design; C: results stem from uncontrolled, nonrandomized, and/or observational studies; D: review or evidence in sufficient for categories A to C.

**Table 2 ijerph-18-13331-t002:** Exercise duration for dementia prevention.

Title	Cohort	Duration	Findings	Highlights	Reference: Grade
Exercise Dose and Aerobic Fitness Response in Alzheimer’s Dementia in the FIT-AD Trial	*n* = 67 median age = 77.4 participants = 26 men and 39 women	1. First training stage at 30 min 2. Gradually lengthened to 60 min 3. 6-month cycling aerobic exercise intervention	aerobic exercise dose was strongly and significantly correlated to change in peak oxygen consumption	Emphasis on exercise dose is needed in aerobic exercise programs to maximize cardiorespiratory fitness gains in persons with mild–moderate Alzheimer’s dementia.	[75]: B
Inter-individual differences in the responses to aerobic exercise in Alzheimer’s disease: Findings from the FIT-AD trial	*n* = 78 age = 77.4 participants = 46 men and 32 women	1. First training stage at 30 min 2. Gradually lengthened to 60 min 3. 6-month aerobic exercise (cycling on recumbent stationary cycles) program	1. Individual differences in aerobic fitness and cognitive responses to aerobic exercise 2. No associations between attendance and changes in aerobic fitness and cognition	The change in aerobic fitness reflects the collective impact of aerobic exercise doses, including frequency/attendance, intensity, session duration, and program duration.	[76]: A
Exercise Is Associated with Reduced Risk for Incident Dementia among Persons 65 Years of Age and Older	*n* = 1740 mean age = 73.2 (free of dementia) mean age = 78.2 (with dementia) mean age = 76.3 (died or withdraw)	at least 15 min (walking, hiking, bicycling, aerobics or calisthenics, swimming, water aerobics, weight training or stretching, or other exercise)	1. Moderate physical activity had a 25% to 38% lower risk for AD 2. Much physical activity had a 33% to 48% lower risk for AD	more physical activity is associated with a reduction in risk for developing AD	[71]: B
A complex multimodal activity intervention to reduce the risk of dementia in mild cognitive impairment–Thinking Fit: pilot and feasibility study for a randomized controlled trial	*n* = 67 mean age = 73.7 participants = 58 men and 9 women	complete a minimum of three, 30–45 min physical activity sessions per week	Significant treatment effects were evident on physical health outcomes, fitness, and cognition.	physical exercises are associated with delayed onset of dementia or Alzheimer’s disease	[77]: B
Resistance Training and Executive Functions: A 12-Month Randomized Controlled Trial	*n* = 155 mean age = 69.6	12 months of progressive resistance training once- or twice-weekly	1. Resistance training groups significantly improved their performance on the Stroop Test 2. Task performance improved in the once-weekly and twice-weekly resistance training groups	Twelve months of once-weekly or twice-weekly resistance training benefited the executive cognitive function of selective attention among senior women.	[78]: A
The impact of resistance exercise on the cognitive function of the elderly	*n* = 62 mean age = 68.15	1. Three one-hour sessions each week 2. 24 wks. of resistance training	1. Experimental moderate presented higher delta means than the control group for the following tests: digit span forward, Corsi’s block-tapping task backward, similarities, and Rey–Osterrieth complex figure immediate recall. 2. Similar for the experimental high group, which showed higher delta means than the control group	Moderate- and high-intensity resistance exercise programs had equally beneficial effects on cognitive functioning.	[79]: B
Aerobic exercise and vascular cognitive impairment: A randomized controlled trial	*n* = 70 mean age = 74	1. Six-month, three-times-weekly 2. Each 60-min class included a 10-min warm-up, a 40-min walk, and a 10-min cool down.	Aerobic exercise training group had significantly improved on Assessment Scale–Cognitive subscale (ADAS-Cog) performance.	a 6-month program of thrice-weekly progressive aerobic training promotes cognitive function and reduces cardiovascular risk in older adults with mild sub-cortical ischemic vascular cognitive impairment (SIVCI)	[80]: A

The level of evidence of each reference is graded as A: overwhelming data from randomized controlled trials (RCTs); B: quasi-experimental design; C: results stem from uncontrolled, nonrandomized, and/or observational studies; D: review or evidence in sufficient for categories A to C.

**Table 3 ijerph-18-13331-t003:** Exercise intensity for dementia prevention.

Title	Cohort	Intensity	Findings	Highlights	Reference: Grade
The effects of aerobic exercise intensity on memory in older adults	*n* = 64 median age = 71.96 participants = 39 men and 25 women	1. Walked four 4-min intervals at 5% grade and 90–95% of peak heart rate/intervals were separated by 3-min active recovery periods (high-intensity interval training group/HIIT) 2. Walking continuously at 70–75% of peak heart rate for 47 min (moderate continuous training group/MCT) 3. Nonaerobic seated and standing stretches group/CON)	1. High intensity interval training(HIIT) induced better memory performance than MCT and CON 2. MCT and CON were not significantly different 3. increases in predicted VO_2_ peak was positively correlated with improvements in high-interference memory performance with a medium effect size	Aerobic exercise may enhance memory in older adults, with the potential for higher intensity exercise to yield the greatest benefit	[84]: B
High-intensity interval exercise improves cognitive performance and reduces matrix metalloproteinases-2 serum levels in persons with multiple sclerosis: A randomized controlled trial	*n* = 60 age = 77.4 participants = 46 men and 32 women	1. Five times of 3-min exercise intervals at 80% of peak oxygen uptake (high-intensity training group/HIIT) 2. 30 min/session at 65% of peak oxygen uptake (control training group/CT)	1. HIIT significantly improved verbal memory 2. HIIT and CT significantly improve in executive functions 3. Significant improvements in VO_2_-peak and a significant reduction in matrix metalloproteinases (MMP)-2 in the HIIT group only	HIIT represents a promising strategy to improve verbal memory and physical fitness in persons with multiple sclerosis.	[86]: A
Aerobic exercise for Alzheimer’s disease: A randomized controlled pilot trial	*n* = 68 mean age = 72.90 participants = 37 men and 61 women	target heart rate (HR) zones from 40–55% to 60–75% of HR reserve	1. When compared to a stretching and toning control intervention, 6-months of aerobic exercise improves functional ability in early-stage AD. 2. Memory performance and brain volume change were linked to gains in cardiorespiratory fitness.	1. Aerobic exercise has been linked to improvements in functional capacity in early AD. 2. Cardiorespiratory fitness gains may be important in driving brain benefits.	[87]: A
Effect of a High-Intensity Exercise Program on Physical Function and Mental Health in Nursing Home Residents with Dementia: An Assessor Blinded Randomized Controlled Trial	*n* = 170 mean age = 86.7 participants = 44 men and 126 women	1. Strengthening exercises should be done for a maximum of 12 repetitions (RM) 2. Control activities: light physical activity, reading, playing games, listening to music and conversations	1. The intervention group outperformed the control group on the Bergs Balance Scale. 2. Significantly improved strength after intervention 3. Level of apathy lower in the exercise group after the intervention	The high intensity functional exercise program improved balance and muscle strength as well as reduced apathy in nursing home patients with dementia.	[88]: A
The Mental Activity and exercise (MAX) Trial A Randomized Controlled Trial to Enhance Cognitive Function in Older Adults	*n* = 126 mean age = 73.4 participants = 47 men and 79 women	12 months of progressive resistance training once- or twice-weekly	1. Over time, the overall cognitive scores improved. 2. Global cognitive scores not differ between groups in the comparison between mental activity intervention group and mental activity control group 3. Comparison between exercise intervention group and exercise control group no difference	The amount of activity is more important than the type in this subject population	[89]: A
Combined Intervention of Physical Activity, Aerobic Exercise, and Cognitive Exercise Intervention to Prevent Cognitive Decline for Patients with Mild Cognitive Impairment: A Randomized Controlled Clinical Study	*n* = 49	target heart rate zone for aerobic exercise during the intervention was at 55–80% of maximum heart rate (HR)	1. The exercise group had significantly better working memory and executive function on the Assessment Scale-Cognitive Subscale (ADAS-Cog). 2. Total physical activity levels were associated with improvements in working memory function and the modified ADAS-Cog score. 3. The associations were stronger for daily moderate intensity activity than for daily step count.	The 24-week combined intervention improved cognitive function and physical function in patients with mild cognitive impairment (MCI) relative to controls.	[90]: A
Dose-Response of Aerobic Exercise on Cognition: A Community-Based, Pilot Randomized Controlled Trial	*n* = 101 mean age = 72.93	In the first 4 weeks of exercise, the target heart rate zone was 40–55% of HRR. In Weeks 5–18, it was 50–65% of HRR. In weeks 19–26, it was 60–75% of HRR.	1. When compared to controls, all exercise groups improved equally in simple attention. 2. A clear dose-response relationship exists between exercise and cardiorespiratory fitness. 3. Cognitive benefits were apparent at low doses with possible increased benefits in visuospatial function at higher doses.	An individual’s cardiorespiratory fitness response was a better predictor of cognitive gains than exercise dose (i.e., duration) and thus maximizing an individual’s cardiorespiratory fitness may be an important therapeutic target for achieving cognitive benefits.	[91]: A

The level of evidence of each reference is graded as A: overwhelming data from randomized controlled trials (RCTs); B: quasi-experimental design; C: results stem from uncontrolled, nonrandomized, and/or observational studies; D: review or evidence in sufficient for categories A to C.

## Data Availability

Not applicable.

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
