# Peer review of "Exercise Dosage in Reducing the Risk of Dementia Development: Mode, Duration, and Intensity—A Narrative Review"

_ijerph, 2021, doi:10.3390/ijerph182413331_

Round 1

Reviewer 1 Report

This is an interesting manuscript dealing with a critical topic. The authors should improve several points though.

Abstract

-Event this is a systematic review, this should include some sentences regarding Methods and Results (databases, number of papers identified). Otherwise, it seems as if you have been cherry picking your studies and thus reaching ad hoc conclusions.

Overall

-The paper should be structured accordingly following the standard points of a scientific study; introduction, methods, results, discussion (including limitations) and conclusion.

-The statements should be made in light of the evidence that you are referring to. For example, if you are citing an observational study, you should be cautious about that, since these types of studies have several risks of bias.

-Academic English should be improved throughout the manuscript. Avoid no formal language.

Introduction

-The definition and the description of the mechanism for developing dementia is fine, but authors should focus more on the link with physical activity

Physical activity/exercise and dementia

-Lines 107-108. Be careful with such statement since there is other type of interventions aside from exercise than could substantially reduce risk for dementia.

-Table 1, Table 2, Table 3 and the grade of evidence should be your results section.

Maintain physical activity and exercise early may prevent Dementia

.Lines 395 and 396. This statement need more explanation or being removed from this section.  

Conclusion

-This is too long and include too specific mention to a specific exercise dosage, which I wouldn´t mention unless you do some metanalysis.

Author Response

Thank you for the review!

Reviewer 1

  1. Event this is a systematic review, this should include some sentences regarding Methods and Results (databases, number of papers identified). Otherwise, it seems as if you have been cherry picking your studies and thus reaching ad hoc conclusions.

A: Thanks for the comment. The purpose of this narrative review manuscript tries to provide an overview of the available research evidence on the benefits of exercise intervention to dementia and describe, discuss the exercise dosage in reducing the risk of dementia development. Please let us provide readers with up-to-date knowledge about this specific topic and theme and reach the goals of narrative review of the literature.

  1. The paper should be structured accordingly following the standard points of a scientific study; introduction, methods, results, discussion (including limitations) and conclusion.

A: Thanks for your comment. Please let this type of narrative review paper use a qualitative approach using the following headings: Introduction, Development (using necessary sub-headings to divide and discuss appropriately the topic), Conclusions, and References that instate of following the standard points of a systematic review paper; introduction, methods, results, discussion (including limitations) and conclusion.

  1. The statements should be made in light of the evidence that you are referring to. For example, if you are citing an observational study, you should be cautious about that, since these types of studies have several risks of bias.

A: Kindly refer to line 284 to line 287. Thank you.

  1. Academic English should be improved throughout the manuscript. Avoid no formal language.

A: We have amended accordingly, thank you.

  1. The definition and the description of the mechanism for developing dementia is fine, but authors should focus more on the link with physical activity/exercise and dementia.

A: Thanks for your comment. Kindly review the Introduction section.

  1. Lines 107-108. Be careful with such statement since there is other type of interventions aside from exercise than could substantially reduce risk for dementia.

A: We have rewritten line 107-108. Thank you.

  1. Table 1, Table 2, Table 3 and the grade of evidence should be your results section.

A: Thanks for your comment. The paper is written in narrative review format; thus, we have Table 1, Table 2, Table 3 in each section to sub-conclude exercise mode, exercise duration and exercise intensity for dementia prevention. Kindly allow the tables to remain in their current place to deliberate the review.

  1. Maintain physical activity and exercise early may prevent Dementia. Lines 395 and 396. This statement needs more explanation or being removed from this section.

A: Line 395 and 396 have been removed from the section. Thank you.

  1. This is too long and include too specific mention to a specific exercise dosage, which I wouldn´t mention unless you do some metanalysis.

A: Thanks for your comment. We have rewritten conclusion and avoid statements that should not appear in the narrative literature review.

Reviewer 2 Report

This review reported the effect of exercise on the risk of dementia. Especially, this narrative review summarized the exercise mode, duration, and intensity in reducing the risk of dementia development. The theme of this review is interesting. However, I have some comments.

 In abstract and conclusions, authors wrote mainly about dementia. Authors should write mainly about the effect of exercise on the risk of dementia development. What kind of exercise does authors recommend? How long and strong is appropriate to reduce the risk? Authors should conclude about the effect of exercise on the risk of dementia based on the narrative review.

 There are many reviews about the relationship between exercise and risk of dementia. What is the different point of this review from the previous reviews?

 Authors wrote in lines 101-102 that “The larger the hippocampus and prefrontal cortex, the larger the hippocampus and prefrontal cortex”. Is this sentence correct? Please confirm it.

 Authors used the abbreviation “PA” (line 113). Does this mean “physical activity”? Authors used the word “physical activity” and “PA” in the text. Please unify the word.

Author Response

Thank you for the review!

Reviewer 2

This review reported the effect of exercise on the risk of dementia. Especially, this narrative review summarized the exercise mode, duration, and intensity in reducing the risk of dementia development. The theme of this review is interesting. However, I have some comments.

  1. In abstract and conclusions, authors wrote mainly about dementia. Authors should write mainly about the effect of exercise on the risk of dementia development. What kind of exercise does authors recommend? How long and strong is appropriate to reduce the risk? Authors should conclude about the effect of exercise on the risk of dementia based on the narrative review.

A: We have amended accordingly in abstract and conclusions. Kindly review these two sections.

  1. There are many reviews about the relationship between exercise and risk of dementia. What is the different point of this review from the previous reviews?

A: Previous reviews has not yet combined exercise mode, exercise duration and exercise intensity for dementia prevention altogether; thus, this review provides a wholesome insight in 3 section at the same time.

  1. Authors wrote in lines 101-102 that “The larger the hippocampus and prefrontal cortex, the larger the hippocampus and prefrontal cortex”. Is this sentence correct? Please confirm it.

A: We have amended accordingly. Thank you.

  1. Authors used the abbreviation “PA” (line 113). Does this mean “physical activity”? Authors used the word “physical activity” and “PA” in the text. Please unify the word.

A: We have unified the word, thank you.

Round 2

Reviewer 1 Report

The authors addressed well my queries. 

Reviewer 2 Report

Most of issues suggested by reviewer were revised correctly. I think this manuscript is now suitable for publication.